# VST-SD: Versatile Style Transfer with Content-Style Statistics Disentanglement

## Abstract

Recent works in versatile style transfer have achieved impressive results in both content preservation and style fidelity. However, optimizing models solely with content and style losses often fails to match the real image distribution, leading to suboptimal stylization quality. In this paper, we propose a novel self-supervised framework, VST-SD, which disentangles content and style representations to enhance stylization performance. Specifically, we separate content and style from the input and train the model to reconstruct the original image. To facilitate effective disentanglement, we leverage feature statistics: a content encoder is designed with perturbation and compression to remove style-related statistics, while a style encoder employs magnitude preservation to capture style-specific information. A cascade of diffusion models are introduced to integrate content and style into new images. To support multi-modal capabilities in versatile style transfer, we construct a paired text-style dataset and design a pipeline enabling flexible, text-guided stylization. Experimental results across artistic, photorealistic, and text-guided stylization demonstrate the effectiveness and versatility of our approach.

## 1 Introduction

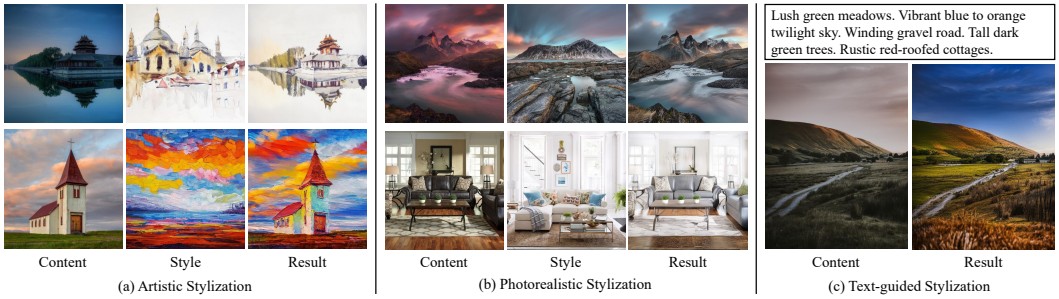

Figure 1: We introduce a self-supervised framework for (a) artistic style transfer and (b) photorealistic style transfer. We further extend it to (c) text-guided style transfer in a unified model.

Style transfer has been widely studied for decades as it provides a convenient way to create artworks and professional photos. It aims to preserve the content of a source image while adopting the style of a reference. In general, content refers to the global structure and semantics, whereas style encompasses visual elements such as texture, color, and high-level appearance attributes (Gatys et al., 2016; Zhang et al., 2023). Recently, versatile style transfer has received a lot of attention (Wen et al., 2023; Huang et al., 2023; He et al., 2025). It extends the traditional style transfer to performs both artistic and photorealistic stylization without retraining or handcrafting separate architecture.

Most existing approaches to versatile style transfer (Li et al., 2019; Hong et al., 2021; Wu et al., 2022; Wen et al., 2023) follow a similar paradigm: they define metrics to quantify content and style similarity (Gatys et al., 2016; Huang & Belongie, 2017), then optimize a model to balance these metrics by preserving content from one image while transferring style from another. While they achieve good balance to produce high similarity results, neither of content and style losses encourage the model to output images that match the real image distributions. As a result, the

stylized images may appear poor in details, inconsistent in texture and color, or overly conservative to avoid visual artifacts. This raises a challenge how to learn real image distributions while achieving high content-style similarity without using predefined similarity losses.

In this paper, we propose VST-SD, a self-supervised framework that bypasses traditional content and style losses. Our approach assumes that if we can successfully disentangle and extract content and style representations from an image, we can train a generator conditioned on these representations to reconstruct the original image. This formulation eliminates the need for handcrafted similarity metrics but introduces the more challenging problem of learning disentangled representations of content and style. While recent diffusion-based disentanglement methods (Wang et al., 2023c; Frenkel et al., 2024; Xing et al., 2024) have shown promise in artistic style transfer, they overfit the style and struggle to preserve image content for versatile style transfer. Previous statistics-based stylization methods (Huang & Belongie, 2017; Risser et al., 2017; Li et al., 2017) report that style can be transferred by minimizing the distance of some channel-wise feature statistics (*e.g.*, mean-std, covariance matrix, and histogram) between the stylized image and reference image. This inspires us to conjecture that channel-wise statistics of original content representation are related to image style. Thus, we leverage feature statistics for disentanglement.

To obtain content representation, we develop a content encoder that suppresses statistical style information. This is achieved by randomly perturbing the statistics of intermediate features and compressing the feature maps along the channel dimension using a lightweight network. For style representation, we design a style encoder that captures feature statistics as style tokens. To mitigate issues related to token magnitude drift, we introduce magnitude preservation to stabilize training.

We train a diffusion model conditioned on the representations to reconstruct the input image. The reconstruction loss encourages the encoders to learn informative representations, while guiding diffusion to learn real image distribution, thus improving both content-style similarity and image quality. To address the problem of losing fine-grained details, we introduce an additional refinement diffusion model, conditioned on the initial stylized output and the style representation.

Moreover, unlike previous methods that rely solely on reference images for style input, our VST-SD supports text-guided stylization (Kwon & Ye, 2022). To address the challenge of data scarcity, we explore vision-language model (Hong et al., 2024) to construct a paired text-style dataset, where each data pair consists of the style tokens extracted from our style encoder and a corresponding style description. A dedicated text-to-style diffusion transformer is trained to generate style tokens from captions. We demonstrate that this text-to-style module can be combined with the content encoder and diffusion, thus achieving a unified framework for both image and text-guided style transfer.

We evaluated VST-SD on a diverse set of images, comparing its image stylization results against versatility methods and disentanglement methods. We also evaluated text-guided stylization through qualitative and quantitative analysis. We demonstrate the benefits of self-supervised framework and the feasibility of achieving image and text-guided stylization in a unified architecture.

Our main contributions are summarized as follows:

- We introduce VST-SD, a self-supervised framework for versatile style transfer based on statistical disentanglement of content and style, supported by a cascade of diffusion models.
- We develop a content encoder that removes style statistics using perturbation and channel-wise compression.
- We design a style encoder with magnitude preservation to robustly capture and stabilize style representations.
- We build a paired text-style dataset and a text-to-style diffusion pipeline, enabling flexible and unified support for both image and text-guided style transfer.

## 2 RELATED WORK

### 2.1 IMAGE STYLE TRANSFER

Artistic style transfer aims to generate image with texture, color from an artwork. Gatys et al. (2016) first explore the generic feature representations of neural networks and introduce neural algorithm

for artistic style transfer. To address the time consuming problem, universal style transfer methods (Huang & Belongie, 2017; Li et al., 2017; Chen & Schmidt, 2016; Sheng et al., 2018; Li et al., 2019) propose efficient feature-space transforms. Subsequent works explore diverse architectures to improve content preservation and style similarity, including transformers (Deng et al., 2022; Tang et al., 2023; Zheng et al., 2024), reversible networks (An et al., 2021; Wen et al., 2023; Liu et al., 2024), and diffusion (Zhang et al., 2023; Deng et al., 2024; Chung et al., 2024; Zhou et al., 2025; Wang et al., 2025).

Photorealistic style transfer has higher demands for content preservation, and stylized images should not be distorted. Luan et al. (2017) first introduce a locally affine transformation with regularization term to enhance photorealism. Following works mainly focus on designing skip connection modules to suppress distortion, or building lightweight networks to improve efficiency. (Li et al., 2018; Yoo et al., 2019; An et al., 2020; Chiu & Gurari, 2022).

Building on these advances, versatile style transfer aims to unify artistic and photorealistic stylization within a single framework. Li et al. (2019) propose a linear transform and post-processing network for artistic and photorealistic stylization, respectively. Hong et al. (2021) introduce domain-aware indicator to adaptively generate artwork or photo. Wu et al. (2022) leverage contrastive learning to improve local coherence. Wen et al. (2023) introduce reversible network with channel refinement module to preserve content affinity.

A fundamental problem in style transfer is to find image representations that independently model the separated content and style. Data-based methods (Zhang et al., 2018; Kotovenko et al., 2019; Wang et al., 2023a) collect images in a pre-defined domains such as characters in different font styles. Recently, disentanglement has been explored in diffusion-based approaches. StyleDiffusion (Wang et al., 2023c) removes style details and fine-tunes a diffusion model (Stability-AI, 2022) to implicitly learn the reference style. B-Lora (Frenkel et al., 2024) finds that the two blocks in SDXL (Podell et al., 2023) individually affect the content and style, and employ LoRA (Hu et al., 2021) to implicitly learn the content and style of an image. CSGO (Xing et al., 2024) improves efficiency by collecting triplet images for training. Unlike these works, which focus primarily on artistic stylization, we achieve disentanglement for versatile style transfer by leveraging feature statistics.

### 2.2 Text-guided Style transfer

Text-guided style transfer extends stylization to natural language prompts. CLIPStyler (Kwon & Ye, 2022) first introduces CLIP matching loss to transfer style from text condition. LDAST (Fu et al., 2022) utilizes annotated artwork datasets to model image-text correlations. ZeCon (Yang et al., 2023) incorporates diffusion models with patch-wise contrastive loss to optimize noise image. Diffstyler (Huang et al., 2024) introduces dual diffusion architecture and learnable noise to control output content. TRTST (Chen et al., 2025) leverages transformer to project text and image into a joint embedding space. In this paper, we leverage vision-language model to learn the correlation between style and text, enabling both image and text-guided stylization in a unified model.

## 3 Method

### 3.1 Framework

Existing versatile style transfer methods typically use a content loss and a style loss (Gatys et al., 2016; Huang & Belongie, 2017) to preserve the content of one image, but the style of another. However, these losses do not encourage the model to approximate the real image distribution, often leading to low-quality or weak stylization. In this paper, we discard the use of content and style loss, but instead propose VST-SD, a self-supervised framework based on content-style disentanglement. Our VST-SD consists of a content encoder $E_c$, a style encoder $E_s$ and a latent diffusion $D$ conditioned on representations from the two encoders. The key to the success of our method is to produce disentangled content and style encoders, with which we can obtain content and style representations from two separate images and then employ the diffusion to interpret them into a new image. To this end, our idea is to extract content representation from a pre-trained network that has some trainable lightweight modules added to help remove the style information in them. For style representation, we build a style encoder and use its feature statistics to construct a set of style tokens. These disentangled representations are then fed into a latent diffusion, which reconstructs the input image by

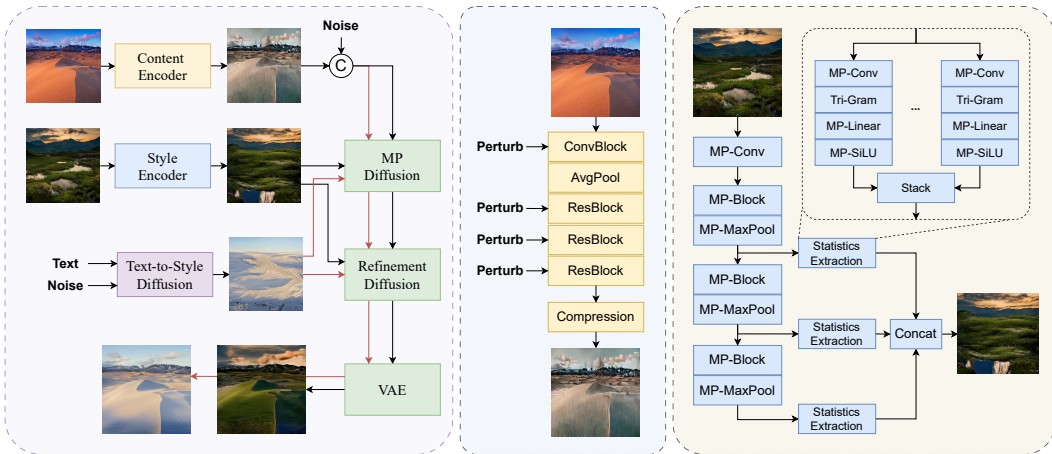

Figure 2: Illustration of VST-SD. (left) Image and text-guided style transfer pipelines. (middle) Compression and perturbation-based content encoder. (right) Magnitude preservation style encoder.

minimizing the following diffusion loss:

$$\mathbb{E}_{\sigma,\phi(I),n}[\lambda(\sigma)\|D(\phi(I) + n; \sigma, E_c(I), E_s(I)) - \phi(I)\|_2^2], \tag{1}$$

where $\sigma$ is the noise level, $\lambda(\cdot)$ is a loss weight, and $\phi$ is the variational autoencoder, $n$ is a random noise $n \sim \mathcal{N}(0, \sigma^2 I)$. To further enhance image details, a refinement diffusion model is further introduced at the end of previous diffusion model.

Finally, to support text-guided stylization, we replace the image-based style encoder branch with a text-to-style diffusion branch while sharing the same content encoder and cascaded diffusion models. Figure 2 illustrates the framework of our method.

## 3.2 STATISTICS-REMOVAL CONTENT ENCODER

Features from pre-trained networks such as VGG or CLIP are rich of high level information and can be used as content representations. However, we find that these features still contain substantial style information. Previous statistics-based methods transfer style by minimizing the distance of some channel-wise statistics between the stylized and reference image's representations (e.g., mean-std in Huang & Belongie (2017), covariance matrix in Li et al. (2017), and histogram in Risser et al. (2017)). This motivates us to explicitly remove channel-wise statistics from pre-trained features to obtain disentangled content representation. In this paper, We employ a CLIP ResNet-101 encoder as the backbone, with five intermediate layers defined at the bottleneck blocks. By visualizing the features at different layers, we observe that the feature from layer 4 captures more global structure and thus use its output as the original content representation. Then, we introduce a trainable lightweight compression module and a perturbation strategy to remove its channel-wise statistics information.

**Channel compression.** Since the statistics relevant to image style is channel-wise, reducing channel dimensionality naturally reduces style information. For example, compressing channels linearly reduces the number of covariance terms quadratically. We thus introduce a lightweight compression module to compress the original content representation along the channel dimension. Here we use consecutive trainable $3 \times 3$ convolution layers to progressively compress the feature maps along the channel dimension. We empirically set the compression ratio to 0.4%, which balances style suppression with content preservation. The spatial dimensions are retained to preserve structure.

**Statistics perturbation.** Distillation method (Chiu & Gurari, 2022) shows that the amount of statistics information can be represented by eigenvectors, and even a single eigenvector can capture more than 12% of original mean-std statistics. Thus, compression alone cannot fully remove them. To further remove statistics information of representation from layer 4, which are the original content representation, we propose to perturb the statistics of output from every layer before it. This perturbation can be realized by using mini-batch statistics and we implement it by reactivating the

batch normalization layers for efficiency. With perturbation, the statistics values of content representation are changed to random values and cannot be recovered. Thus, the diffusion generator is forced to synthesize image using statistics information from the style representation. Together, compression and perturbation significantly improve disentanglement between content and style.

### 3.3 MAGNITUDE-PRESERVATION STYLE ENCODER

While the content encoder preserves global structure, the style encoder must capture low-level information. Our style encoder is designed with magnitude preservation layers to stabilize training. It consists of multiple resolution levels, with deeper blocks at lower resolutions. The self-attention layer is inserted at low resolution to obtain large receptive field. We extract the features at different resolution, where the last one has the same downsampling ratio as the content representation.

**Statistics extraction.** At each resolution, we flatten intermediate feature and compute Gram matrix (Gatys et al., 2016). Then, we flatten the lower triangular elements into vector. This yields a set of multi-scale tokens as style representation. Although capturing different scale statistics, our experiment finds that the result is still of low style similarity. We consider that the capacity of style information is limited. Thus we improve it by splitting each feature into $N$ groups along the channel, and process each group with a mapping network. This produces multi-head style tokens $S$:

$$S = \{s_i^l | i \in \{1, 2, ..., N\}, l \in \{1, 2, ..., L\}\}, \tag{2}$$

where $s_i^l$ is the $i^{th}$ style token in layer $l$. To keep the capacity balance between style encoder and diffusion, we set $N = 4$ by default.

**Magnitude preservation.** During training, the magnitude of style tokens can grow excessively (over $10^{10}$ times). Since style is related to statistics, we avoid feature normalization layers. Instead, we propose to preserve the output magnitude of each layer. For trainable layers, Karras et al. (2020; 2024) show that normalizing layer weights has the same effect as normalizing features in terms of constraining magnitude. We thus apply normalization on the weights of each trainable layer. For non-parametric layers, we compute the ratios between the channel-wise mean of the output and the input from a set of samples before training, and then use the corresponding ratio to rescale the output of each layer. These strategies effectively stabilize training without degrading style information.

### 3.4 CONTENT-STYLE CONDITIONED CASCADE OF DIFFUSION

**Magnitude preservation diffusion.** For diffusion model, we adopt the UNet architecture. To unify the learning rate, the diffusion is designed with magnitude preservation layers as style encoder. To integrate content representation, we concatenate it with the noise along the channel dimension. To inject style information, we add cross-attention layers to transform the channel correlations of each feature element. Each residual block is followed by a self-attention layer and a cross-attention layer, except at the highest and lowest resolutions. To reduce computation, images are compressed into latent space using a pretrained VAE (Rombach et al., 2022). The intermediate content feature is resized via nearest interpolation before compression.

**Refinement diffusion.** The images generated by the first stage diffusion may lack fine details, especially in photorealistic stylization. To address the problem, we introduce a second stage refinement diffusion conditioned on noise image and style. We initialize it from SDXL refiner weights (Stability-AI, 2022) and add additional learnable cross-attention layers to enable conditioning on style tokens. During inference, we first sample a stylized latent from the first stage diffusion, then refine it over first 100 noise scales of second stage diffusion. Figure 3 shows that the refinement diffusion improves the image quality.

### 3.5 TEXT-TO-STYLE DIFFUSION

Existing versatile style transfer requires preparing style image in advance, which are may inaccessible in practical situations. To enable flexible stylization, we introduce a text-to-style diffusion pipeline. Due to lack of training data, we started by constructing a dataset of text-style pairs. Since vision-language model (VLM) shows remarkable capabilities in image understanding (Hong et al.,

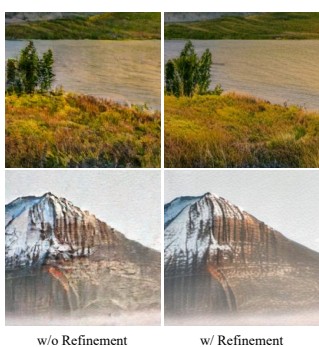

w/o Refinement    w/ Refinement

Figure 3: Ablation results of refinement diffusion.

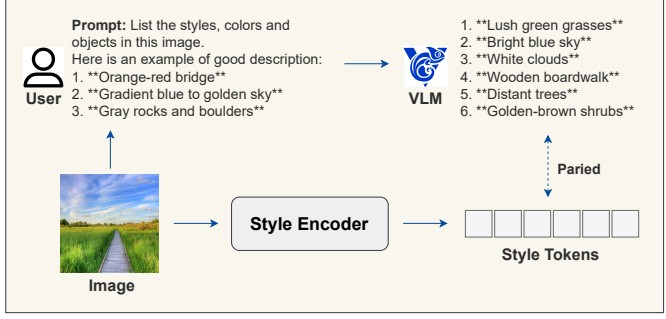

Figure 4: The pipeline for paired text-style data generation. We utilize VLM and style encoder to construct text-style pairs.

2024), we propose to employ VLM to generate image captions in term of styles, colors, and objects. To address the issue of caption redundancy, we force the VLM to generate captions in a specific format by giving an example, which is also convenient for user to input at test time. Meanwhile, we also use the style encoder to extract style tokens for the images. Figure 4 shows the data generation process. We finally train a text-to-style diffusion transformer (Peebles & Xie, 2023) on the generated dataset where each data sample consists of a style description and a sets of style tokens. At inference, the generated tokens from text-to-style diffusion are fed into the cascaded diffusion pipeline, replacing the style tokens output from the image-based style encoder. This enables flexible and unified support for both image and text-guided stylization.

## 4 EXPERIMENTS

### 4.1 IMPLEMENT DETAILS

For image style transfer, we train on images from the LAION-Aesthetics dataset (LAION-AI, 2022), filtering approximately 10 million images with aesthetics scores above 6. All images are resized and randomly cropped to $256 \times 256$. For text-guided style transfer, we construct 100k paired text–style samples from the same dataset. We adopt exponential moving averaging (Karras et al., 2023) and train the model on NVIDIA A100 GPUs with a batch size of 96. Mixed precision training with bf16 is employed, and we use Adam with a learning rate of 2e-3. For diffusion sampling, we use deterministic sampling with 32 steps. In terms of guidance, we dropout the content-style representations and text for unconditional image and text-to-style generation, respectively. The classifier-free guidance weight is set to 1.6 by default.

### 4.2 IMAGE STYLE TRANSFER

We compare our method against versatile style transfer models (Hong et al., 2021; Wu et al., 2022; Wen et al., 2023), which are trained with content and style losses, and disentanglement-based methods (Ahn et al., 2024; Frenkel et al., 2024; Xing et al., 2024) that build upon pretrained Stable Diffusion (Stability-AI, 2022). For photorealistic style transfer, we randomly select 100 photo images for both content and style. For artistic style transfer, we additionally select 100 artistic images as style. These generate 20,000 stylized images for evaluation.

**Qualitative results.** Figure 5 shows the results generated by different methods. Versatility methods (Hong et al., 2021; Wu et al., 2022; Wen et al., 2023) are able to generate high style similarity. However, they typically generate images with poor details since optimizing content and style losses cannot guarantee image quality. Disentanglement methods (Ahn et al., 2024; Frenkel et al., 2024; Xing et al., 2024) generate clean and smooth images. However, DreamStyler and B-LoRA fail to preserve image structure and semantics, while CSGO compromises on style similarity. Compared with above methods, our VST-SD not only generates high quality stylized images, but also preserve content structure and semantics.

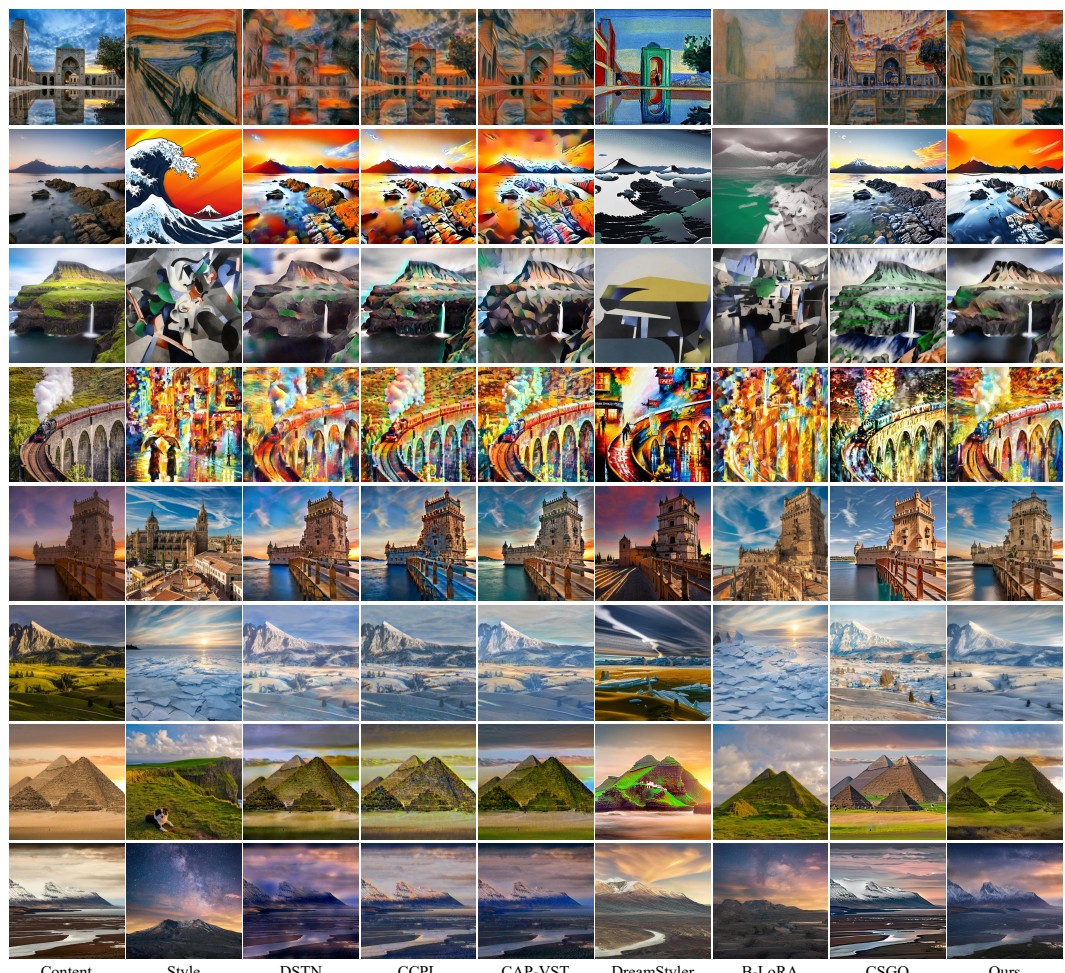

Content  Style  DSTN  CCPL  CAP-VST  DreamStyler  B-LoRA  CSGO  Ours

Figure 5: Qualitative comparison of versatile style transfer. The first four rows shows artistic style transfer, while the last four rows show photorealistic style transfer.

Table 1: Quantitative comparison of versatile style transfer. The two values represent results for artistic style transfer and photorealistic style transfer, respectively. Execution time is reported for a resolution of 512×512.

| | Metrics | $L_{Gram} \downarrow$ | $S_{CLIP} \uparrow$ | $D_{Style} \downarrow$ | CLIP-IQA↑ | Time (s)↓ |
|---|---|---|---|---|---|---|
| Versatility Methods | DSTN (Hong et al., 2021) | 2.75/1.12 | 0.73/0.72 | 0.75/0.32 | 0.46/0.83 | 0.54 |
| | CCPL (Wu et al., 2022) | 1.04/0.92 | 0.69/0.68 | 0.32/0.30 | 0.57/0.69 | 0.12 |
| | CAP-VST (Wen et al., 2023) | 1.20/0.36 | 0.67/0.71 | 0.39/0.10 | 0.50/0.91 | **0.10** |
| Disentangle-ment Methods | DreamStyler (Ahn et al., 2024) | 7.92/4.31 | 0.78/0.79 | 1.71/0.91 | 0.54/0.92 | 959.80 |
| | B-LoRA (Frenkel et al., 2024) | 2.89/2.31 | **0.80/0.80** | 0.58/0.45 | 0.57/0.90 | 1138.78 |
| | CSGO (Xing et al., 2024) | 6.02/4.52 | 0.74/0.73 | 1.54/1.22 | **0.64**/0.90 | 21.95 |
| | Ours | **0.94/0.28** | 0.75/0.76 | **0.24/0.07** | 0.63/**0.93** | 4.80 |

**Quantitative results.** we evaluate the style similarity and the image quality. For the metrics of style similarity, the Gram loss ($L_{Gram}$) (Gatys et al., 2016) measures the low-level style (e.g., texture and color), and CLIP similarity ($S_{CLIP}$) measures the high-level style (e.g., semantic elements and object shape). To assess overall similarity, we report the style distance $D_{style} = L_{Gram}(1 - S_{CLIP})$. For the metrics of image quality, we adopt the CLIP Image Quality Assessment (CLIP-IQA) (Wang et al., 2023b). Table 1 shows the quantitative results. Versatility methods (Hong et al., 2021; Wu et al., 2022; Wen et al., 2023) train the models with style loss, thus achieving high style similarity. However, the stylized image is of low quality as content and style losses cannot match real image

distribution. Disentanglement methods (Ahn et al., 2024; Frenkel et al., 2024; Xing et al., 2024) successfully capture high-level style and achieve better CLIP similarity, but this comes at the cost of content preservation. Besides, they are unable to capture low-level style. These problems can be attributed to text-to-image models capturing only high-level semantics. In contrast, our VST-SD achieves good performance in both style similarity and image quality. Even without training on style loss, VST-SD achieves a lower style distance than the versatility method, which demonstrates the effectiveness of the proposed disentanglement.

For inference speed, versatility methods are an order of magnitude faster than diffusion-based disentanglement methods. VST-SD outperforms other disentanglement methods as it directly performs diffusion sampling.

### 4.3 Text-guided Style Transfer

We present comparison of VST-SD against open-source models, including CLIPstyler (Kwon & Ye, 2022), ZeCon (Yang et al., 2023), and Diffstyler (Huang et al., 2024). For evaluation, we use 786 photo images from (Xia et al., 2020) as content. To obtain style text, we first use VLM to generate a textual description (Figure 4), then we ask VLM to produce a variation of the text containing the same object but different style and color.

**Qualitative results.** Figure 6 shows the comparison of text-guided style transfer results. CLIPstyler (Kwon & Ye, 2022) optimizes image with CLIP loss, which produces noticeable artifacts and results in outputs that do not resemble a real image. ZeCon (Yang et al., 2023) optimizes noise from diffusion with CLIP loss. However, it often adds new elements and cannot preserve image content. Diffstyler (Huang et al., 2024) only preserves the outline and cannot preserve semantics. Compared with above methods, our VST-SD generates faithful stylization which align with the text and match the distribution of real images.

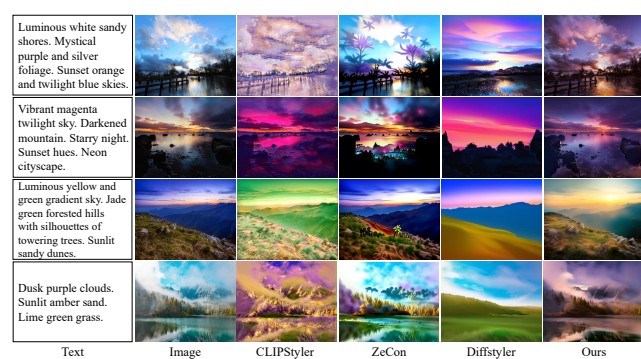

Figure 6: Qualitative comparison of text-guided style transfer.

Table 2: Quantitative comparison of text-guided style transfer.

|  | Text Alignment↑ | Image Quality↑ | Time (s)↓ |
|---|---|---|---|
| CLIPstyler | **0.33** | 0.87 | 78.01 |
| ZeCon | 0.30 | 0.92 | 52.92 |
| Diffstyler | 0.30 | 0.79 | 24.10 |
| Ours | 0.28 | **0.93** | **7.67** |

**Quantitative results.** we use CLIP similarity to asses the text alignment, and CLIP-IQA (Wang et al., 2023b) to measure the image quality. Table 2 shows the quantitative results. CLIPstyler (Kwon & Ye, 2022), ZeCon (Yang et al., 2023), and Diffstyler (Huang et al., 2024) achieve high text alignment as they directly optimizes with CLIP loss. However, they produces poor image quality. In contrast, our VST-SD aligns text without altering the image content, and achieves high image quality. Additionally, VST-SD is faster at inference since it directly performs image sampling.

### 4.4 Analysis

#### 4.4.1 Content and Style Representations

To investigate what is encoded in the learned content and style representations, we invert them back to image space with our first stage diffusion. Specially, we feed the content representation into diffusion model along with a random noise as style representation, which generates inverted content $I_c$. For style representation, we feed it into diffusion model along with a zero tensor eliminating structure and semantics as content representation, which generates inverted style $I_s$. Figure 7 shows the visualization examples. The inverted content $I_c$ maintains the global structure while removing other information related to style. On the contrary, the inverted style $I_s$ presents similar texture and color while being agnostic to structure. Interestingly, $I_c$ and $I_s$ are semantically entangled. It can be explained by content representation as the superconcept and style representation as the subconcept.

Figure 7: Visualization of the learned content and style representations. The inverted content $I_c$ preserves the global structure of original image, while the inverted style $I_s$ presents similar texture and color.

Table 3: Ablation results for the content encoder (left) and style encoder (right).

| Channel compression | Statistics perturbation | $L_{Gram}\downarrow$ | $S_{CLIP}\uparrow$ | $D_{Style}\downarrow$ |
|---|---|---|---|---|
| – | – | 2.04/0.66 | 0.69/0.73 | 0.64/0.18 |
| ✓ | – | 1.31/0.29 | 0.73/0.74 | 0.36/0.07 |
| – | ✓ | 1.40/0.39 | 0.71/0.74 | 0.40/0.10 |
| ✓ | ✓ | **0.97/0.23** | **0.75/0.75** | **0.24/0.06** |

| Statistics extraction | Magnitude preservation | $L_{Gram}\downarrow$ | $S_{CLIP}\uparrow$ | $D_{Style}\downarrow$ |
|---|---|---|---|---|
| – | – | 2.21/0.42 | 0.72/0.72 | 0.61/0.12 |
| ✓ | – | 1.65/0.27 | 0.74/0.71 | 0.43/0.08 |
| – | ✓ | 1.21/0.34 | 0.74/0.74 | 0.32/0.09 |
| ✓ | ✓ | **0.97/0.23** | **0.75/0.75** | **0.24/0.06** |

From photorealistic style transfer (Luan et al., 2017), semantical entanglement helps model to render style to semantically related regions and generate consistent stylization of the same semantic region.

### 4.4.2 ABLATION

We conduct ablation study to evaluate the proposed components in content encoder and style encoder with first stage diffusion. Table 3 show the ablation results. Both channel compression and statistics perturbation helps to remove style information from content representation, thus achieving high style similarity. Replacing statistics extraction with the Gram matrix in (Gatys et al., 2016), the information of style representation is limited. With magnitude preservation layers, the model learns more meaningful style representation. When all components are used, the model achieves the highest style similarity.

### 4.4.3 MODEL SCALE

To investigate the impact of model scale, we train first stage diffusion of varying training images and parameters. Quantitative evaluation is shown in Table 4. We can see that using larger dataset Laion-Aesthetics leads to improvement in style similarity, and increasing model parameters improve the image quality. We find the scaling trend shows no signs of saturation, which makes us optimistic about continuing to improve model performance with more parameters and training images.

Table 4: Quantitative analysis of model scale.

| Dataset | Trainable param. | $L_{Gram}\downarrow$ | $S_{CLIP}\uparrow$ | $D_{Style}\downarrow$ | CLIP-IQA$\uparrow$ |
|---|---|---|---|---|---|
| MS-COCO & WikiArt | 178M | 1.13/0.24 | 0.75/0.74 | 0.28/0.06 | **0.60**/0.89 |
| Laion-Aesthetics | 178M | 0.97/0.23 | 0.75/0.75 | 0.24/0.06 | 0.57/0.92 |
| Laion-Aesthetics | 667M | **0.67/0.21** | **0.76/0.76** | **0.16/0.05** | 0.59/**0.92** |

## 5 CONCLUSION

In this paper, we presented VST-SD, a self-supervised framework for versatile style transfer based on feature statistics disentanglement. We demonstrated that channel-wise statistics largely encode style information and proposed compression and perturbation strategies to remove them from the content representation. To complement this, we designed a magnitude-preserving style encoder that effectively captures style statistics without instability. These disentangled representations are integrated within a scalable diffusion model, whose performance consistently improves with larger model capacity and training data. Furthermore, we extended the framework to text-guided style transfer by leveraging vision–language models and the style encoder to establish explicit correlations between text and style tokens, enabling more flexible and practical applications. Extensive experiments confirm that VST-SD achieves superior results both qualitatively and quantitatively compared to state-of-the-art approaches. In future work, we aim to extend our framework to video style transfer, which presents additional challenges in temporal consistency and computational efficiency.

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

## A  ADDITIONAL ANALYSIS

To further investigate the relationship between the input image, the content representation, and the style representation, we conduct quantitative evaluations using 10k randomly selected images $I$. For each image, we generate the inverted content $I_c$ and the inverted style $I_s$, and then compute the Gram loss and CLIP cosine similarity. As shown in Table 5 (a), $I_c$ and $I_s$ are effectively disentangled in terms of low-level texture and color, while $I$ and $I_s$ share similar style. Table 5 (b) shows that $I_c$ and $I_s$ are entangled at high-level semantics, which helps model to generate consistent stylization of the same semantic region.

We also analyze the feature spaces of the content encoder and style encoder. Specifically, we compute the mean squared error between the features of $I$, $I_c$, and $I_s$ using each encoder. The results, shown in Table 6, reveal that images with similar structures are closer in the content encoder's feature space, whereas images with similar styles are closer in the style encoder's feature space. This validates that our disentanglement strategy successfully assigns structure and style information to the appropriate representations.

Table 5: Quantitative analysis of the learned content and style representations. $I$: original image, $I_c$: inverted content, $I_s$: inverted style.

| (a) $L_{Gram}$ | $I$ | $I_c$ | $I_s$ | (b) $S_{CLIP}$ | $I$ | $I_c$ | $I_s$ |
|---|---|---|---|---|---|---|---|
| $I$ | 0.00 | 2.94 | 0.26 | $I$ | 1.00 | 0.84 | 0.77 |
| $I_c$ | 2.94 | 0.00 | 2.78 | $I_c$ | 0.84 | 1.00 | 0.81 |
| $I_s$ | 0.26 | 2.78 | 0.00 | $I_s$ | 0.77 | 0.81 | 1.00 |

Table 6: Quantitative analysis of content and style encoder feature space. $I$: original image, $I_c$: inverted content, $I_s$: inverted style.

| (a) Content distance | $I$ | $I_c$ | $I_s$ | (b) Style distance | $I$ | $I_c$ | $I_s$ |
|---|---|---|---|---|---|---|---|
| $I$ | 0.00 | 0.08 | 0.37 | $I$ | 0.00 | 0.30 | 0.01 |
| $I_c$ | 0.08 | 0.00 | 0.42 | $I_c$ | 0.30 | 0.00 | 0.31 |
| $I_s$ | 0.37 | 0.42 | 0.00 | $I_s$ | 0.01 | 0.31 | 0.00 |

## B  STYLE INTERPOLATION

We further explore the flexibility of our model by interpolating between different style representations. Given a style representation $S_c$ extracted from a content image and a style representation $S_s$ obtained from a reference image or text, we perform linear interpolation as follows:

$$S_{mix} = (1 - \alpha)S_c + \alpha S_s, \tag{3}$$

where $\alpha \in [0, 1]$ controls the interpolation ratio between the two styles. Figures 8 and 9 illustrate that gradually varying $\alpha$ produces smooth transitions from one style to another, demonstrating that our disentangled style representation is well-suited for controllable style blending.

## C  REFINEMENT DIFFUSION

The refinement diffusion is designed to enhance fine-grained details in the images generated by the magnitude-preservation diffusion stage. We initialize it from the SDXL refiner (Stability-AI, 2022) and augment it with an additional style cross-attention layer to explicitly incorporate style information. Specifically, we insert the style cross-attention layer immediately after the text cross-attention layer, and use the same query $Q$ for both operations. The output of the combined cross-attention is formulated as:

$$Z_{out} = \text{Attention}(Q, K_t, V_t) + \text{Attention}(Q, K_s, V_s), \tag{4}$$

where $K_t, V_t$ denote the key and value derived from text features, and $K_s, V_s$ represent the key and value derived from style features. The text prompt is set to "professional, highly detailed" by default.

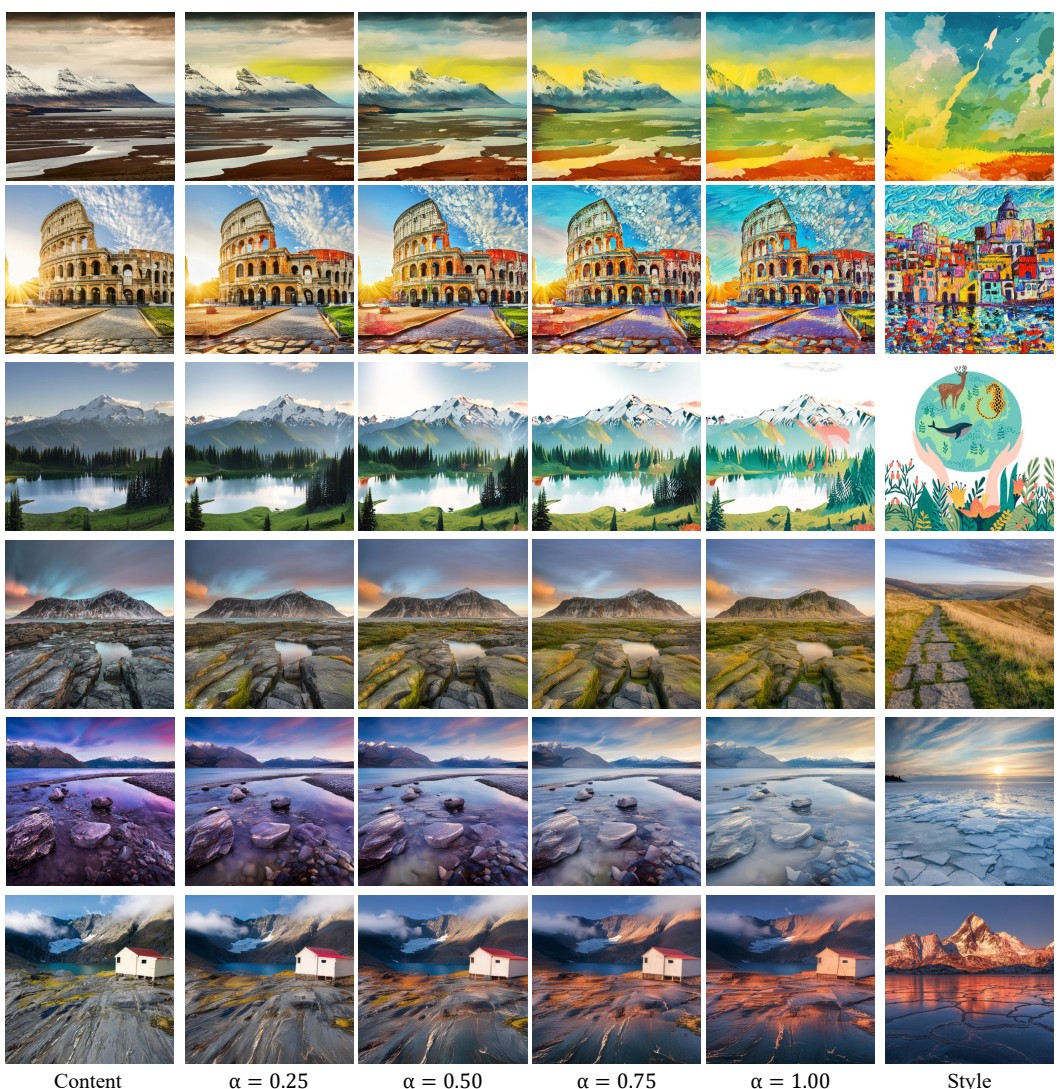

|Content|α = 0.25|α = 0.50|α = 0.75|α = 1.00|Style|

Figure 8: Style interpolation. The first three rows shows artistic style transfer, while the last three rows show photorealistic style transfer.

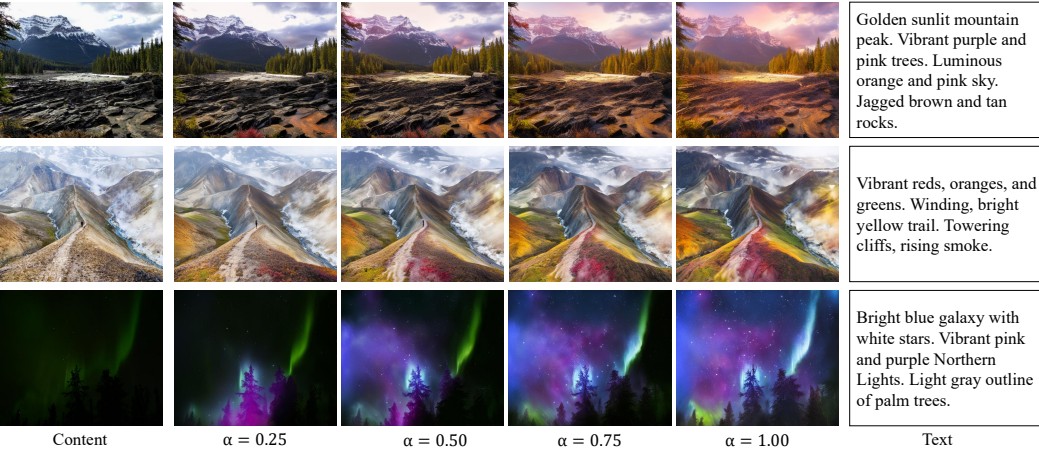

|Content|α = 0.25|α = 0.50|α = 0.75|α = 1.00|Text|

Figure 9: Style interpolation on text-guided style transfer.

| Model | Content encoder | Style encoder | MP diffusion | Refinement diffusion | Text-to-style diffusion |
|---|---|---|---|---|---|
| Resolution | 256×256 | 256×256 | 32×32 | 64×64 | - |
| Channels | 256 | 512 | 256 | 384 | 1024 |
| Channel multiplier | 1,2,4 | 1,1,1,1,1 | 1,2,3,4 | 1,2,4,4 | 1 |
| Blocks | 3,4,23 | 3,3,4,4,8 | 4,4,4,4 | 2,2,2,2 | 24 |
| Attention resolutions | - | 16,32 | 8,16 | 16,32 | - |
| Head channels | - | 64,64 | 64,64 | 64,64 | 64 |
| Attention type | - | self | self+cross | self+cross | self+cross |
| Context dim | - | - | 512 | 1280,512 | 2048 |
| EMA power | 2/3 | 2/3 | 2/3 | - | 2/3 |
| Warm-up iterations | 5000 | 5000 | 5000 | 5000 | 5000 |
| Trainable param. | 10.6M | 64.5M | 592.0M | 32.1M | 559.3M |
| Non-trainable param. | 27.6M | 0.0M | 0.0M | 2314.8M | 0.0M |
| Total param. | 38.2M | 64.5M | 592.0M | 2346.8M | 559.3M |

Table 7: Hyperparameters for model training.

## D TEXT-TO-STYLE DIFFUSION

Our text-to-style diffusion adopts the diffusion transformer as the base architecture. Each transformer block consists of a multi-head self-attention layer and a feed-forward layer. A multi-head cross-attention layer is placed after the self-attention layer to interact with the text embedding. Following the EDM formulation (Karras et al., 2022), we implement the denoiser $D_{text2style}$ that predicts denoised style tokens as:

$$D_{text2style}(z; \sigma, c) = c_{skip}(\sigma)z + c_{out}(\sigma)F(c_{in}(\sigma)z; c_{noise}(\sigma), c), \quad (5)$$

where $z$ is the noisy style tokens, $\sigma$ is the noise level, $c$ is the text condition, and $F$ denotes the transformer network. $c_{in}$, $c_{noise}$ $c_{skip}$ and $c_{out}$ are the precondition terms to keep input and output signal magnitudes fixed. Since EDM (Karras et al., 2022) has strong constraints on the mean-variance of data, we normalize the input style tokens with predefined values which is computed by randomly select a set of style tokens from dataset and calculate the mean-variance. For noise level $\sigma$, we use log-normal distribution, i.e. $\ln(\sigma) \sim \mathcal{N}(P_{mean}, P_{std}^2)$ with $P_{mean} = -1.2$ and $P_{std} = 1.2$. To enable classifier-free guidance, the text is randomly dropout $10\%$ of the time during training.

## E HYPERPARAMETERS

Table 7 lists the configuration of each network in the model during training.

## F ADDITIONAL COMPARISONS

Figure 10 and 11 show additional artistic and photorealistic style transfer qualitative comparison results with versatility methods and disentanglement methods.

Figure 12 shows additional text-guided style transfer qualitative comparison results.

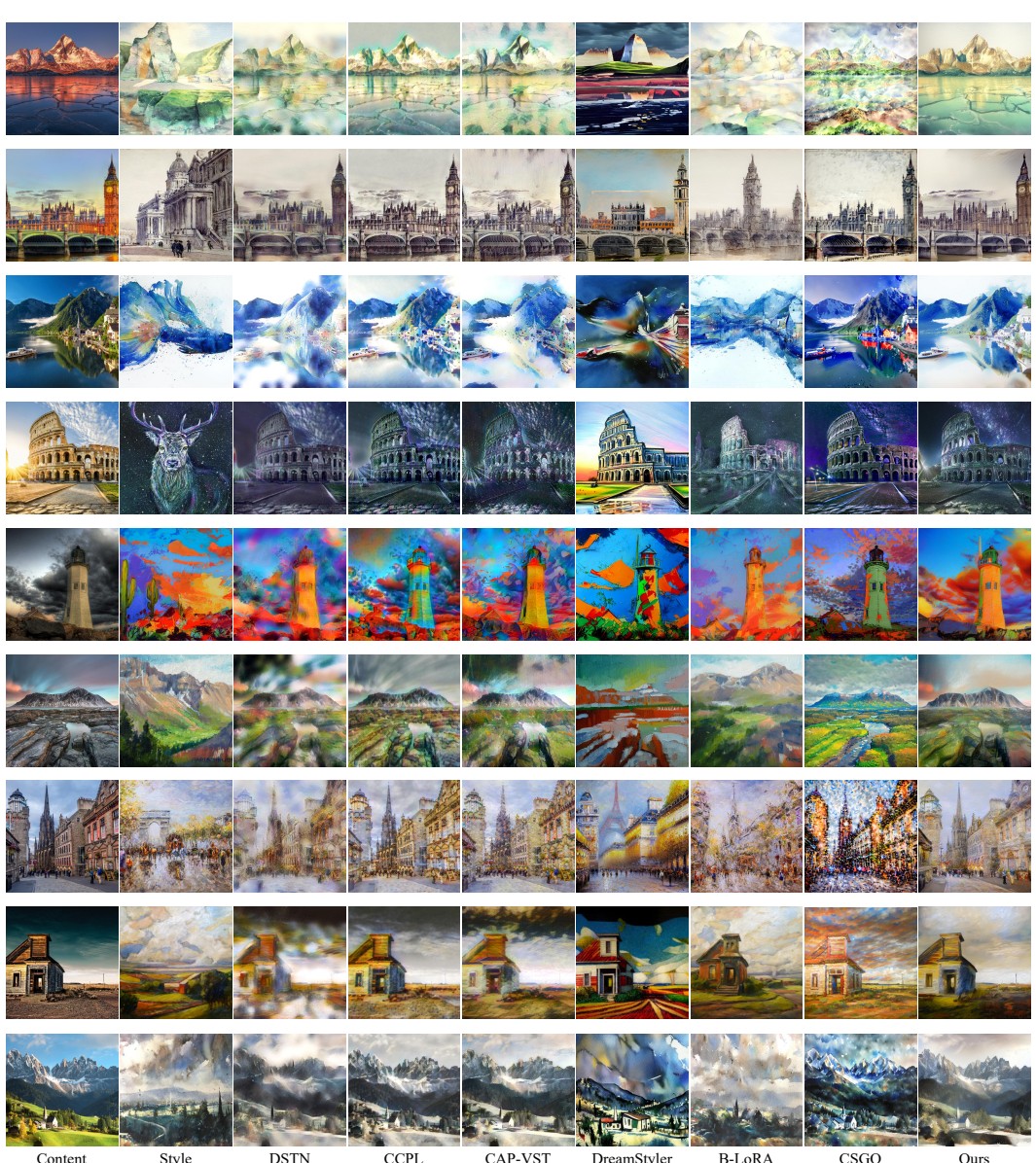

Content    Style    DSTN    CCPL    CAP-VST    DreamStyler    B-LoRA    CSGO    Ours

Figure 10: Qualitative comparison of artistic style transfer.

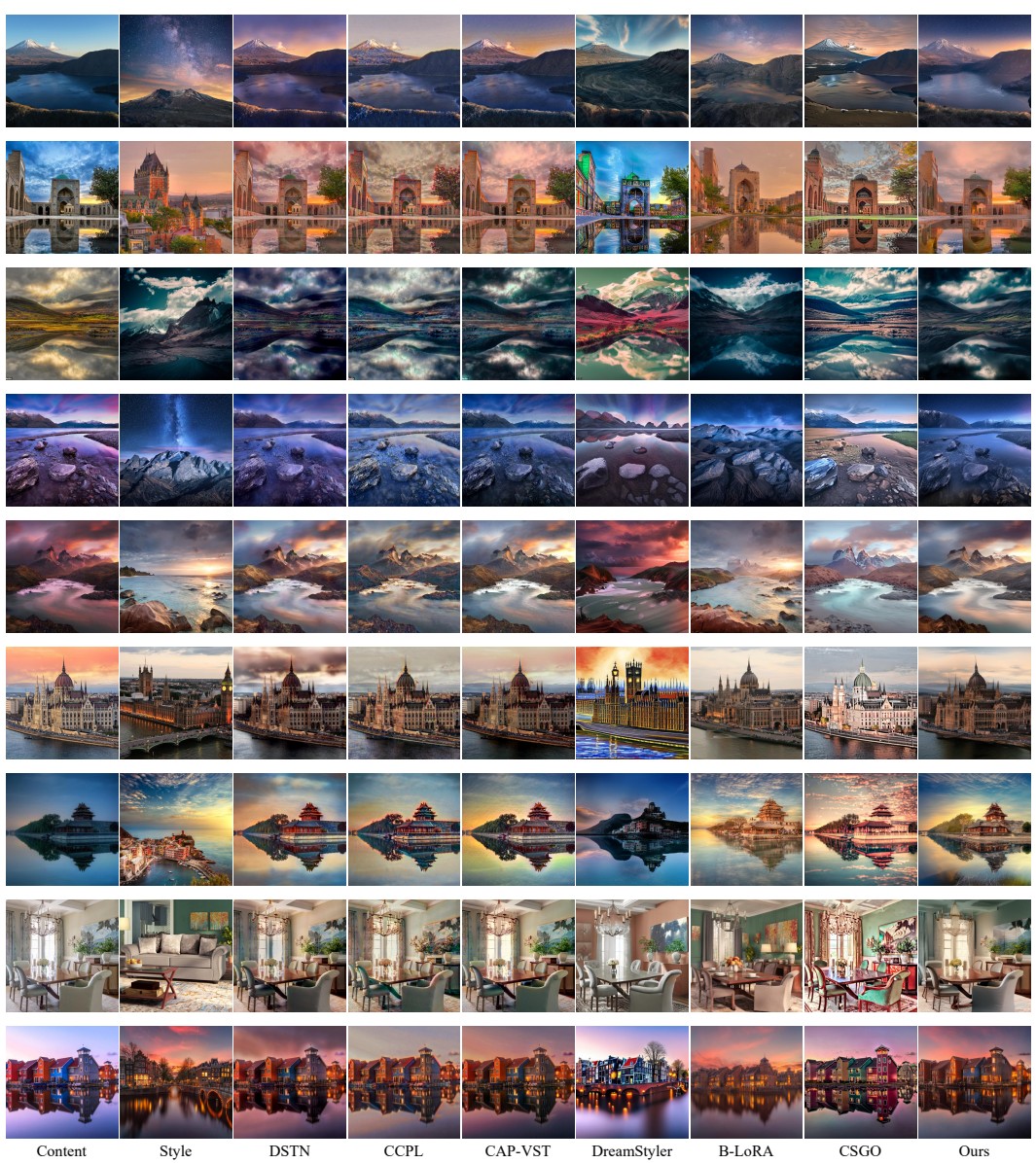

Figure 11: Qualitative comparison of photorealistic style transfer.

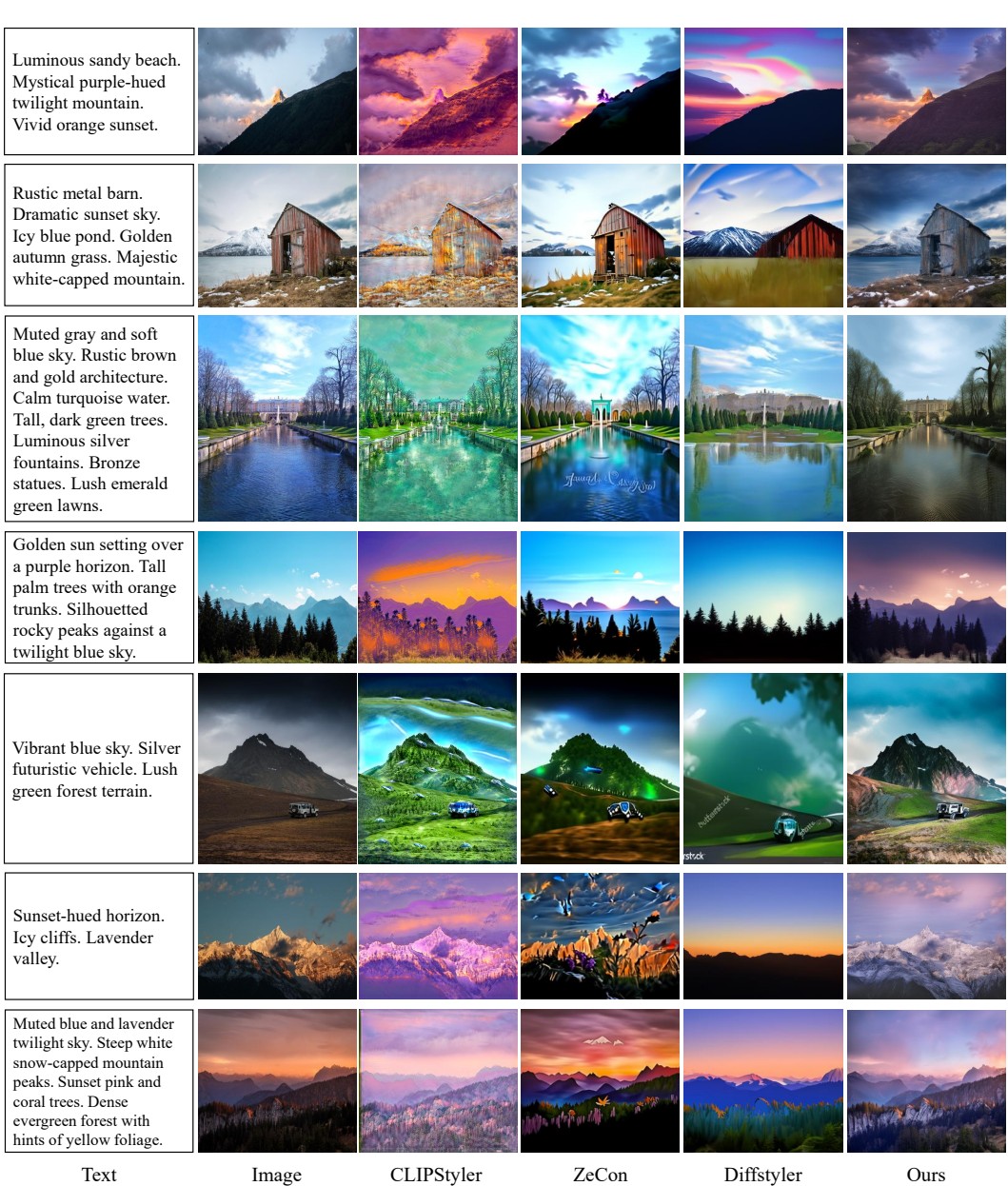

Figure 12: Qualitative comparison of text-guided style transfer.

