# OpenReview forum: "VST-SD: Versatile Style Transfer with Content-Style Statistics Disentanglement"
_ICLR.cc/2026/Conference — ICLR 2026 Conference Withdrawn Submission_

### Official Review · Reviewer_CtAs · 2025-10-19

**Soundness:** 2
**Presentation:** 1
**Contribution:** 1
**Rating:** 2
**Confidence:** 5

**Summary:**

This paper presents VST-SD, a diffusion-based versatile stylization method leveraging feature statistics disentanglement. This method involves a self-supervised learning framework that trains a diffusion model to reconstruct from the disentangled content and style representations. Traditional statistical methods rely on auxiliary objectives to learn the disentanglement of style and content features, while suffering from poor image quality. Instead, the authors propose to directly learn the disentangled representations in an unsupervised manner, and leverage the diffusion generative model to learn the stylized image distribution. Additionally, the authors introduce a diffusion-transformer to generate style representations from text to further enable flexible text-guided image stylization. This generation pipeline is a cascade of three separate diffusion models: 1) a main stylized image generator; 2) a refiner initialized with the first stage coarse generation; 3) a text-to-style generator to generate style representations from text.

**Strengths:**

- The proposed method seems to be lightweight and fast.
- The design of using diffusion models for text-to-style representation token generation, instead of directly extracting text representation, is novel.
- The presented qualitative samples are high-quality with plausible style transfer. The quantitative comparisons demonstrate the advantages of the proposed method.

**Weaknesses:**

- Legend is missing in Figure 2. The red line in the left chart is confusing, does it represent split operation?
- The overall experimental results seem to be sparse. Specifically, the quantitative evaluations are conducted on 100 images sampled from the training set. Moreover, only photorealistic stylization is tested in Section 4.3, I would like to see how the proposed method behave on other styles like digital art etc.. I believe more evaluation results on a public benchmark like IMAGStyle [1] will better showcase the effectiveness of the proposed method.
- In addition to the sparsity in test data, the employed evaluation metrics are also constrained. There are four quantitative metrics reported, three of which evaluate the style consistency and the last one is CLIP-IQA which evaluates image quality. I believe there should be some metrics measuring the content similarity as well. For instance, leveraging the proposed content encoder for such evaluation will not only demonstrate the effectiveness of the encoder itself, but also be a good quantitative metric for comparing content similarity.
- Besides the limitation in quantitative metrics, the proposed method fails on the CLIP-based style similarity evaluation. Notably, both of the other two style-consistency metrics are related with Gram Loss. I wonder if these two advantageous metrics are correlated with each other, and in that case the superiority of the proposed method would be much unclear. Overall, I believe there should be more and better designed evaluations.
- The analysis of the entanglement of style and content representation in Section 4.4.1 is interesting. The authors implement a smart way to visualize the extracted representations. But if the inverted content and style are entangled, it is super unclear how the proposed method works since it is built on the assumption of ‘a generator conditioned on the disentangled content and style representations can reconstruct the original image’ (line 58-60).
- The ablation setting is ambiguous. Are the reported results in Table 3 evaluated on separate models trained w/ and w/o the listed components? Or did the author directly evaluate the model performance without statistical perturbation, for instance, even though the model was trained with it. Moreover, since the visualization analysis in 4.4.1 couldn’t demonstrate the disentanglement effect. Maybe another set of ablation studies on the compression rate in Section 3.2 can help us figure out how the encoder learns the disentanglement of style and content features?
- Finally, limitations of the proposed method are not discussed.

[1] Csgo: Content-style composition in text-to-image generation. arXiv preprint arXiv:2408.16766, 2024.

**Questions:**

In line 199-201, the authors said ‘By visualizing the features at different layers, we observe that the feature from layer 4 captures more global structure’. Can we have some visualized samples presented here?

**Details Of Ethics Concerns:**

I believe there is no special ethics concerns in this paper.

---

### Official Review · Reviewer_4tVJ · 2025-10-22

**Soundness:** 2
**Presentation:** 3
**Contribution:** 2
**Rating:** 4
**Confidence:** 4

**Summary:**

This paper proposes a new versatile style transfer framework called VST-SD, which can perform artistic, photorealistic, and text-guided stylization in a unified model. VST-SD consists of two key components: a statistics-removal content encoder that removes style statistics using perturbation and channel-wise compression, and a magnitude-preservation style encoder that robustly captures and stabilizes style representations.

**Strengths:**

1. The proposed method supports three kinds of style transfer: artistic, photorealistic, and text-guided style transfer, in a unified model.
2. The images generated by this method for artistic and photorealistic style transfer are impressive.
3. Extensive experiments are conducted to evaluate the performance of the proposed method.

**Weaknesses:**

1. The proposed method shows unsatisfactory performance in text-guided style transfer. For example, in the first row of Figure 4, the description in the text instruction, "Luminous white sandy shores. Sunset orange and twilight blue skies" is not reflected in the generated image. The quantitative results in Table 2 also demonstrate that the proposed method performs the worst in text alignment compared to all other baseline methods.

2. In Section 3.2, the paper states, 'By visualizing the features at different layers, we observe that the feature from layer 4 captures more global structure and thus use its output as the original content representation,' but no specific experimental results are provided to support this conclusion.

3. In the ablation experiment shown in Figure 3, the difference between using and not using refinement diffusion is not significant. In this case, it is recommended to provide quantitative experiments to further demonstrate the effectiveness of the refinement diffusion.

4. In the ablation experiment in Section 4.4.2, there is a lack of qualitative experimental results, which are the most direct way to observe the effects of each component.

5. Some state-of-the-art style transfer methods are not introduced and compared in this paper, such as SaMam [1], HIS [2], OmniStyle [3], and StyleSSP [4]. \
[1] SaMam: Style-aware State Space Model for Arbitrary Image Style Transfer. CVPR 2025. \
[2] HSI: A Holistic Style Injector for Arbitrary Style Transfer. CVPR 2025. \
[3] OmniStyle: Filtering High Quality Style Transfer Data at Scale. CVPR 2025. \
[4] StyleSSP: Sampling StartPoint Enhancement for Training-free Diffusion-based Method for Style Transfer. CVPR 2025.

**Questions:**

Please see **Weaknesses**.

Others:

1. Section 3.2 states, "This perturbation can be realized by using mini-batch statistics, and we implement it by reactivating the batch normalization layers for efficiency." Why does reactivating the batch normalization layers help achieve perturbation?

2. Section 3.3 states, "we flatten the lower triangular elements into a vector." Why were the lower triangular elements chosen? What is the purpose of doing this?

---

### Official Review · Reviewer_pSmz · 2025-10-28

**Soundness:** 2
**Presentation:** 2
**Contribution:** 2
**Rating:** 4
**Confidence:** 4

**Summary:**

This paper presents a self-supervised framework for image/text-based style transfer. The core innovation lies in its approach to content-style disentanglement by leveraging feature statistics. The authors' motivation for designing the content-style disentanglement method stems from previous research on the relationship between image style and feature statistics. By employing channel-wise compression and perturbation, they encourage the disentanglement of style information. Through comparative experiments, the authors claim that their method achieves state-of-the-art performance in style transfer.

**Strengths:**

I appreciate the paper's motivation to abandon traditional style loss and adopt a self-supervised approach for style disentanglement. Style transfer has long lacked convincing evaluation metrics, making it necessary to revisit this task from fundamental principles. Additionally, the paper achieves not only image-guided style transfer but also text-guided style transfer.

**Weaknesses:**

- While numerous prior studies have demonstrated the effectiveness of style transfer by manipulating statistical features, they have not proven that style information strictly corresponds to these statistics. Therefore, using this as a self-supervised objective cannot achieve complete disentanglement of style information. As shown in Figure 7, the content image still retains abstract stylistic attributes from the original image, such as brushstrokes.

- The experimental details are insufficient—for instance, implementations and hyperparameters of compression and perturbation, as well as the construction details of the text-style dataset (Figure 4 appears to only illustrate object or object-color pair).

- The refinement diffusion is the most computationally heavy component in the pipeline, yet its necessity lacks further discussion. There are no quantitative metrics provided, and the qualitative results in Figure 3 show almost no discernible difference.

**Questions:**

Please address the concerns raised in the weaknesses section.

---

### Official Review · Reviewer_WkK1 · 2025-10-30

**Soundness:** 3
**Presentation:** 3
**Contribution:** 3
**Rating:** 6
**Confidence:** 3

**Summary:**

This paper proposes a novel self-supervised framework named VST-SD for Versatile Style Transfer (VST), which unifies artistic style transfer, photorealistic style transfer, and text-guided style transfer. Experimental results demonstrate that the proposed method outperforms existing approaches on both qualitative and quantitative metrics across all three tasks.

**Strengths:**

1. The method achieves strong performance across three distinct style transfer tasks.
2. The paper's core contributions statistics removal in the content encoder and magnitude preservation in the style encoder are key to achieving high-quality self-supervised disentanglement. This approach of explicitly disentangling statistics is more fundamental and effective than traditional methods that rely on predefined loss functions.

**Weaknesses:**

1. The paper states that statistics perturbation is achieved by reactivating the batch normalization layers. This is unclear. Does this simply mean using training mode during inference? If so, this is just standard batch normalization, not a novel perturbation.
2. The paper lacks a discussion on abstract style texts. The text-to-style module appears to be limited to descriptive styles . This is an important limitation that the paper does not discuss.
3. Is "Dstulet" in Table 3 a typo for "Dstyle"?

**Questions:**

1. The paper mentions that "statistical perturbation is achieved by reactivating the batch normalization layers", but the mechanism is not clearly described. Does "reactivating" here specifically refer to using the training mode of batch normalization during inference? If it is only this operation, what is the difference between it and standard batch normalization, and how to reflect the innovation of "perturbation"?​
2. The existing research does not involve discussions on abstract style texts, and the text-to-style module only shows adaptability to descriptive styles. Could you supplement the explanation of the module's limitations in processing abstract style texts, and whether there are optimization directions for this limitation in the future?
3. The term "Dstulet" in Table 3 is different from the context and the common term "Dstyle". Is "Dstulet" a typo of "Dstyle"?

---

### Note · Authors · 2025-11-12

**Comment:**

Thanks the reviewers for taking the time to review this paper.

I have read and agree with the venue's withdrawal policy on behalf of myself and my co-authors.

**Withdrawal Confirmation:**

I have read and agree with the venue's withdrawal policy on behalf of myself and my co-authors.